# Impaired Butyrate Induced Regulation of T Cell Surface Expression of CTLA-4 in Patients with Ulcerative Colitis

**DOI:** 10.3390/ijms22063084

**Published:** 2021-03-17

**Authors:** Maria K. Magnusson, Alexander Vidal, Lujain Maasfeh, Stefan Isaksson, Rajneesh Malhotra, Henric K. Olsson, Lena Öhman

**Affiliations:** 1Department of Microbiology and Immunology, Institute of Biomedicine, Sahlgrenska Academy, University of Gothenburg, 405 30 Gothenburg, Sweden; maria.magnusson@microbio.gu.se (M.K.M.); lujain.maasfeh@gu.se (L.M.); stefan.isaksson@microbio.gu.se (S.I.); 2Bioscience In Vivo, Research and Early Development, Respiratory & Immunology, BioPharmaceuticals R&D, AstraZeneca, 405 30 Gothenburg, Sweden; alexander.vidal@astrazeneca.com; 3Translational Science and Experimental Medicine, Research and Early Development, Respiratory & Immunology, BioPharmaceuticals R&D, AstraZeneca, 405 30 Gothenburg, Sweden; rajneesh.malhotra@astrazeneca.com (R.M.); henric.k.olsson@astrazeneca.com (H.K.O.)

**Keywords:** ulcerative colitis, short-chain fatty acids, butyrate, T cells, CTLA-4

## Abstract

Patients with ulcerative colitis (UC) have reduced intestinal levels of short-chain fatty acids (SCFAs), including butyrate, which are important regulators of host–microbiota crosstalk. The aim was therefore to determine effects of butyrate on blood and intestinal T cells from patients with active UC. T cells from UC patients and healthy subjects were polyclonally stimulated together with SCFAs and proliferation, activation, cytokine secretion, and surface expression of cytotoxic T lymphocyte-associated antigen-4 (CTLA-4) were analyzed. Butyrate induced comparable, dose dependent inhibition of activation and proliferation in blood T cells and activation in intestinal T cells from UC patients and healthy subjects. However, surface expression of the inhibitory molecule CTLA-4 on stimulated blood and intestinal T cells was impaired in UC patients and was not restored following butyrate treatment. Furthermore, unlike intestinal T cells from healthy subjects, butyrate was unable to downregulate secretion of interferon gamma (IFNγ), interleukin (IL)-4, IL-17A, and IL-10 in UC patients. Although seemingly normal inhibitory effects on T cell activation and proliferation, butyrate has an impaired ability to reduce cytokine secretion and induce surface expression of CTLA-4 in T cells from UC patients with active disease. Overall, these observations indicate a dysfunction in butyrate induced immune regulation linked to CTLA-4 signaling in T cells from UC patients during a flare.

## 1. Introduction

Inflammatory bowel disease (IBD), mainly comprising ulcerative colitis (UC) and Crohn’s disease (CD), is a chronic intestinal disorder ascribed to an exaggerated and protracted immune activity directed against the commensal microbiota which eventually results in inflammation of the intestine [1,2]. The pathophysiology of the disease is incompletely understood but involves genetic, environmental and microbial factors. In UC, the inflammation involves the superficial mucosal layer, it extends proximally from rectum into the colon in a continuous fashion and affected individuals experience diarrhea, rectal bleeding, and abdominal pain [2]. Although malfunctions in innate immunity seem to pave the way for inflammation, a key role for dysregulated T cell responses in propagation and perpetuation of the consequent inflammation in the gut is widely accepted [3].

In the healthy gut, the host-microbiota interplay is a crucial regulatory factor, training the immune system to differentiate the commensal bacteria from pathogens and thus sustaining homeostasis. This crosstalk is largely achieved through byproducts of microbial metabolism of dietary components such as short chain fatty acids (SCFAs), primarily butyrate, propionate and acetate [4]. These metabolites exert their effects by binding to G-protein-coupled receptors and/or inhibiting histone deacetylases (HDACs) [5]. The immunoregulatory effect of SCFAs on immune cells in the gut may be of importance for IBD. For instance, butyrate has been reported to inhibit NFKB signaling and decrease pro-inflammatory cytokine production in blood and intestinal mononuclear cells from patients with CD [6]. Butyrate has also been demonstrated to decrease expression of Th17 cytokines while inducing differentiation of blood T regulatory cells (Treg) in a rat model of colitis [7].

T cell activation and proliferation is downregulated by cytotoxic T lymphocyte-associated antigen-4 (CTLA-4) binding to the costimulatory molecules CD80 and CD86 on antigen presenting cells [8]. While CTLA-4 is not expressed by resting T cells, its expression on the cell surface is transiently upregulated upon activation, as a regulatory negative feedback mechanism [9], and immunoregulation of T cell responses through CTLA-4 is thus important for maintaining gut homeostasis. In healthy mice, blocking of CTLA-4 for a limited time induces spontaneous autoimmune disease [10]. Further, the most common side effect of anti-CTLA-4 immunotherapy in cancer patients is enterocolitis, most likely due to increased intestinal levels of the pro-inflammatory cytokines IFNγ and IL-17A [11,12].

Although SCFAs have been recognized as potential therapeutic agents for IBD, clinical attempts have not always been rewarding [13,14,15,16]. Therefore, clarification of the effects of SCFAs, and especially butyrate, on T cell activity in the context of IBD is needed. The aim of this study was therefore to determine effects of butyrate on blood and intestinal T cells in UC patients. 

## 2. Results

### 2.1. SCFAs Inhibit Blood T Cell Proliferation and Activation in Healthy Subjects and Patients with Active Ulcerative Colitis

To evaluate the effects of SCFAs on CD4^+^ and CD8^+^ T cell proliferation and activation, peripheral blood mononuclear cells (PBMCs) were polyclonally stimulated together with increasing concentrations (0–6.4 mM) of butyrate, proprionate, and acetate. Median frequencies of live cells (live/dead aqua^−^ or 7AAD^−^) and apoptotic cells (AnnexinV^+^) were 87–89% and 0.5–1.5%, respectively, except for cell cultures with 6.4 mM butyrate where median frequency of live and apoptotic cells were 44% and 7.5%, respectively (Appendix A). Therefore, proprionate and acetate concentrations of 0–6.4 mM and butyrate concentrations of 0–1.6 mM were used for subsequent studies.

The effects of SCFAs on CD4^+^ and CD8^+^ T cell proliferation and activation were first analyzed in blood cells from healthy subjects (Figure 1a). A dose-dependent suppression of CD4^+^ and CD8^+^ T cell proliferation by addition of butyrate and proprionate, but not acetate, was detected (Figure 1b). T cell activation, determined by median fluorescent intensity of CD25 expression on CD4^+^ and CD8^+^ T cells, was dose-dependently suppressed by butyrate and proprionate, while no effect was recorded by the addition of acetate to the cell cultures (Figure 1c). Similarly, frequencies of CD25^+^ cells among total CD4^+^ and CD8^+^ T cells were reduced by butyrate and proprionate, but not acetate (Appendix A). Also, in patients with active UC median fluorescent intensity of CD25 expression on CD4^+^ and CD8^+^ T cells was found to be suppressed by butyrate (Figure 1d).

To verify involvement of HDAC class I and II inhibition, purified CD4^+^ T cells were stimulated for 48 h and subsequently treated with increasing doses of SCFAs and analyzed for HDAC I and II activity. Most prominent suppression was detected for butyrate followed by proprionate, while the effects of acetate were more modest (Figure 1e). Due to this, cells were treated only with butyrate for the remainder of the study.

### 2.2. Butyrate Induces Surface Expression of CTLA-4 on Stimulated Blood T Cells from Healthy Subjects but not on Blood T Cells from Patients with Active Ulcerative Colitis

Activation of T cells is accompanied by recruitment of the inhibitory receptor CTLA-4 to the surface of T cells for regulation. Surface expression of CTLA-4 on both CD4^+^ and CD8^+^ blood T cells after polyclonal stimulation was determined (Figure 2a). Butyrate concentrations of 0.1 mM and 0.4 mM induced CTLA-4 expression on proliferating CD4^+^ and CD8^+^ T cells in healthy subjects while no effects were detected for patients with active UC (Figure 2b). In addition, CTLA-4 expression was higher on both CD4^+^ and CD8^+^ T cells in healthy subjects than UC patients, both with and without addition of butyrate (0.1–0.4 mM) (Figure 2b). In summary, T cell surface expression of CTLA-4 expression is reduced in UC patients and is not restored by butyrate treatment.

### 2.3. Butyrate Inhibits Lamina Propria T Cell Activation in Non-Inflamed Subjects and Patients with Active Ulcerative Colitis

Next, we analyzed the effects of butyrate on lamina propria (LP) T cell activation. Sigmoidal LP T cells from UC patients with active disease and non-inflamed subjects were polyclonally stimulated with increasing concentrations of butyrate. Activation, determined by median fluorescent intensity of CD25 expression, was efficiently reduced for CD4^+^ and CD8^+^ T cells in both study groups (Figure 3). Similar results were obtained when comparing stimulated LP T cells from non-inflamed ascending colon from the same study subjects as above (UC patients and non-inflamed subjects) (Appendix A).

### 2.4. Impaired Butyrate Induced CTLA-4 Expression and Altered Cytokine Profile of Lamina Propria T Cells from Patients with Active Ulcerative Colitis

Surface expression of CTLA-4 on CD4^+^ and CD8^+^ LP T cells after polyclonal stimulation was determined (Figure 4a). The addition of 0.4 mM butyrate induced CTLA-4 expression on sigmoidal CD4^+^ LP T cells in both non-inflamed subjects and patients with active UC while no effect was detected for CD8^+^ LP T cells (Figure 4b). Further, CTLA-4 expression was higher on both CD4^+^ and CD8^+^ T cells in non-inflamed subjects as compared to UC patients with active inflammation (Figure 4a,b). A similar pattern was detected in LP T cells from non-inflamed ascending colon from the same study subjects as above (UC patients and non-inflamed subjects) (Appendix A).

Next, we analyzed cytokine expression in the supernatants from stimulated sigmoidal LP cells. For non-inflamed subjects, IFNγ, IL-4, IL-17A, and IL-10 were decreased by addition of butyrate, while no effects were detected for patients with active UC (Figure 5).

## 3. Discussion

The current study demonstrates that although butyrate equally inhibits activation and proliferation of blood and intestinal T cells obtained from UC patients and healthy subjects, butyrate does not downregulate cytokine secretion of intestinal T cells recovered from UC patients. Further, the surface expression of CTLA-4 on stimulated T cells from UC patients is reduced and not restored by butyrate treatment. Thus, our data suggest an impaired butyrate induced control of T cell function linked to CTLA-4 in patients with UC, potentially resulting in dysregulated immune activity. 

Similar to previous reports, treatment with butyrate and proprionate, but not acetate, dose-dependently decreased the proliferation and activation of T cells [17,18,19,20]. This is probably due to higher HDAC inhibition potency of butyrate and proprionate compared to acetate, although an insufficient acetate concentration cannot be ruled out as an explanation for the dissimilarity in effect of SCFAs on T cells. While SCFAs also can act through the G-protein coupled receptors GPR41 and GPR43, this is less likely the signaling pathway of importance, as T cells do not express these receptors at a functional level [21].

While alleles linked to stability and expression levels of CTLA-4 have been described in UC, the functional role in disease pathogenesis is yet to be established [22]. Our study demonstrates that, similar to blood T cells, intestinal T cells from UC patients had lower expression of CTLA-4 following polyclonal activation compared to that of healthy subjects, regardless of whether the UC biopsies were inflamed or not. Our findings are supported by Wang et al. (2019), demonstrating that the expression of CTLA-4 is lower in follicular regulatory T cells from UC patients than in T cells from healthy individuals [23]. Moreover, butyrate upregulates CTLA-4 expression on blood T cells from healthy subjects, but fails to do so in UC patients. For intestinal T cells, CTLA-4 is induced on CD4^+^ but not CD8^+^ T cells, both for healthy individuals and UC patients. The discrepancy between CTLA-4 regulation at different sites may be inherent to the fact that the majority of the blood T cells are naïve while the intestinal T cells are effector T cells, influenced by the local environment. CTLA-4 expression and function can be suppressed in T helper cell 17 (Th17) promoting milieu [24], a known condition of UC patients [25,26]. Thus, a dysregulated Th17 response might have contributed to the impaired CTLA-4 expression capacity of T cells from UC patients in our study. Moreover, while intestinal T cells of healthy subjects and UC patients secreted comparable cytokine levels after polyclonal stimulation, butyrate treatment decreased cytokine secretion of intestinal T cells from healthy subjects but not from UC patients, indicating a higher resistance to the immune modulating effect of butyrate in UC. 

High levels of butyrate and proprionate have been reported to interfere with the anti-tumor activity of CTLA-4 blockade in cancer patients and associates to higher frequencies of blood Tregs [27]. In our setting, we have not specifically studied Tregs and the polyclonal stimulation used does not promote their formation even though many of the T cells, especially from the intestinal tissue, are likely to be Tregs. CTLA-4, a molecule important for negative feedback mechanisms of T cell activity, is known to be highly expressed on Tregs but is also expressed on conventional T cells during activation. The extent of immune suppression will be dependent on the expression level of CTLA-4 both for intrinsic and extrinsic regulation. The intrinsic regulation includes disruption of intracellular T cell receptor signaling inhibiting T cell activation upon binding to co-stimulatory molecules [8] as well as CTLA-4-induced T cell motility limiting the interaction time between the T cell and the antigen presenting cell [28]. The extrinsic regulation includes blocking and internalization of co-stimulatory molecules [29]. Together these act both on the innate and adaptive arms of the immune system to dampen the inflammatory response.

IBD is commonly associated with altered gut microbiota composition, including decreased abundance of SCFA producing bacteria like *Roseburia hominis* and *Faecalibacterium prausnitzii* resulting in lower levels of butyrate, which has been hypothesized to play a role in the etiology of the disease [30,31]. Although there have been attempts to increase the gut butyrate levels in IBD patients both through SCFA promoting high-fiber diets or rectal administration of butyrate enemas, therapeutic outcomes have not met initial expectations [14,32]. In addition to decreased intestinal butyrate levels, an impaired utilization of butyrate in UC patients has been proposed [33,34,35]. Further, UC patients with mild to severe disease activity have been shown to express low levels of genes involved in butyrate uptake and oxidation [36]. Also, a recent pathway enrichment analysis showed altered butyrate metabolism in UC patients [37], and our group demonstrated that butyrate more potently down-regulates gene expression of inflammatory pathways in non-inflamed controls than in inflamed tissue of UC patients [38]. It should be noted that the effect of butyrate may differ according to the local microenvironment. For instance, butyrate, which otherwise promotes epithelial barrier integrity, could be detrimental in the presence of tumor necrosis factor alpha and IFNγ [39]. Butyrate can also induce different T cell responses depending on the cytokine milieu [21].

There are limitations of this study, and we acknowledge the small sample size as a caveat of the current study and the need for validation in a larger cohort for more grounded conclusions. Also, histological evaluation of intestinal tissue as well as location of different T cell subsets in the biopsies would have been optimal; however, all available biopsies were used to obtain enough cells for the butyrate analysis. Moreover, profiling the T cells prior to stimulation and butyrate treatment may have provided a better understanding of effects of butyrate on specific T cell subsets. Furthermore, although not part of this study, and the effect size is probably small, the potential indirect effects of butyrate via other cells in the cultures should also be considered. On the other hand, this study was not limited to blood T cells and included intestinal T cells from inflamed and non-inflamed colon of UC patients and healthy subjects. The results obtained from blood and intestinal T cells were consistent, which strengthens the validity and reliability of the current conclusions. Finally, while CTLA-4 expression was only assessed extracellularly, it is known that the majority of CTLA-4 is localized intracellularly and cycles between intracellular vesicles and the cell surface upon activation [40]. Nevertheless, CTLA-4 can only assert its functions when cycled to the cell surface, therefore only extracellular expression was evaluated.

To conclude, although butyrate inhibits activation and proliferation of blood and intestinal T cells obtained from both UC patients and healthy subjects, butyrate does not downregulate cytokine secretion of intestinal T cells recovered from UC patients. Also, T cell surface expression of CTLA-4 expression is reduced in UC patients and is not restored by butyrate treatment. Overall, these observations indicate a dysfunction in butyrate induced T cell regulation linked to CTLA-4 signaling for patients with UC.

## 4. Materials and Methods

### 4.1. Demographics of Participating Subjects and Sample Collection

Study subjects were recruited among patients at the outpatient clinic at the Sahlgrenska University Hospital, Gothenburg, Sweden and among healthy volunteers at AstraZeneca, Mölndal, Sweden. Venous blood was obtained from nine healthy subjects (four females) between 18–65 years although exact age of donors was hidden to the study. Biopsies were obtained from six non-inflamed subjects (four females, mean age 53, range 22–77) undergoing colonoscopy for other indications than inflammation (polyps, weight loss). Additionally, venous blood was obtained from seven UC patients (two females, mean age 55, range 32–78) and colonic biopsies from six UC patients (two females, mean age 47, range 40–58). The patients with UC had active disease (endoscopic Mayo score 3; *n* = 1, 2; *n* = 8, 1; *n* = 2) and a median disease duration of 18 years (0–41). Current treatments were 5-ASA (*n* = 7), cortison (*n* = 1), thiopurines (*n* = 1), 5-ASA/thiopurines (*n* = 2) or 5-ASA/anti-TNF (*n* = 1), one patient was without treatment.

### 4.2. Blood and Lamina Propria Cell Isolation

Peripheral blood mononuclear cells (PBMCs) were isolated from heparinized venous blood by density-gradient centrifugation on Ficoll-Paque (GE Healthcare, Danderyd, Sweden). For studies of histone deacetylase (HDAC) I and II activity, CD4^+^ T cells were purified from PBMCs using Dynabeads^TM^ CD4 (Invitrogen, Carlsbad, CA, USA), according to the manufacturer’s manual.

Biopsies were collected in PBS and put on ice. Epithelial cells were removed by incubating for 15 min at 37 °C with HBSS-EDTA (HBSS containing 2% FCS, 1.5 mM Hepes and 2 mM EDTA) three times followed by a wash in RPMI 1640 containing 10% FCS and 1.5 mM Hepes (all from Sigma-Aldrich, Saint Louis, MO, USA). Lamina propria (LP) cells were prepared by a 45–60 min incubation at 37 °C with 2 mg/mL collagenase D (Sigma-Aldrich) and 60 U/mL DNase I (Sigma-Aldrich) diluted in RPMI 1640 with 10% FCS, 1.5 mM Hepes and 2 mM CaCl_2_ (Sigma-Aldrich) and filtered through a nylon mesh.

### 4.3. Cell Cultivation Assays

Total PBMC or LP cell populations (2 × 10^5^ cell/well) were cultured for 3 or 2 days, respectively, at 37 °C and 5% CO_2_, in Iscove’s medium supplemented with 10% human FBS, 100 µg/mL gentamicin (Sigma-Aldrich) and 3 µg/mL L-glutamine (Sigma-Aldrich), without or with different concentrations of sodium acetate (S2889), sodium proprionate (P1880) or sodium butyrate (B5887, all from Sigma Aldrich). The cell cultures were stimulated polyclonally in flat-bottomed 96-well plates (Nunc, Roskilde, Denmark). Plates were precoated with 3 µg/mL goat anti-mouse IgG (Jackson Immunoresearch labs, West Grove, PA, USA), and 12.5 ng/mL mouse anti-CD3mAb (BD Pharmingen, San Diego, CA, USA) and 0.05 µg/mL soluble anti-CD28mAb (BD Pharmingen) were added together with the cells. T cell proliferation was measured by dilution of 5, 6-carboxyfluorescein diacetate succinimidyl ester (CFSE) dye. CFSE (Invitrogen) was added to 1 × 10^6^ cells/mL PBMCs in a final concentration of 0.2 µM and incubated at 37 °C for 10 min prior to polyclonal stimulation. Cells were immediately washed in Iscove’s medium supplemented with 10% human FBS, 100 µg/mL gentamicin and 3 µg/mL L-glutamine after the CFSE incubation period. Proliferating cells were defined as cells that had undergone at least one cell division using non-stimulated cells as control. 

### 4.4. Histone Deacetylase (HDAC) I and II Measurement

CD4^+^ T cells were stimulated as described above, but only for 48 h, and were then detached, washed once, counted, and added to 384-well plates at a density of 1000 cells/well. Cells were then stimulated with anti-CD3 and anti-CD28 (as above), with and without addition of butyrate, proprionate or acetate and incubated at 37 °C and 5% CO_2_ for 1.5 h. Relative activity of HDAC I and II enzymes was measured using the HDAC-Glo™ I/II Assay and Screening System (Promega, Madison, WI, USA) according to the manufacturer’s instructions.

### 4.5. Flow Cytometry

Cultured cells were stained for flow cytometry analysis of various surface receptors using combinations of the following antibodies: anti-CD3-BV786 or BV605, anti-CD4-PeCy7 or PerCP-Cy5.5, anti-CD8-FITC or APC-H7, anti-CD25-BV421, anti-CD25-PE and mouse IgG2aκ-PE isotype control (all from BD Biosciences, San Jose, CA, USA). Live/Dead Fixable Aqua Dead Cell Stain Kit (Molecular Probes, Invitrogen) or 7-Aminoactinomycin D (7AAD, BD Biosciences) were used to exclude non-viable cells according to the manufacturers’ protocols. For analysis of apoptosis, cells were labelled with Annexin V diluted in Annexin V Binding Buffer (BD Biosciences). Samples were collected with a LSRII or FACSFortessa X20 flow cytometer (BD Biosciences) using DIVA software (BD Biosciences) and analyzed using FlowJo software (Tree Star Inc., Ashland, KY, USA). Gates were set using Fluorescence Minus One (FMO) or isotype control.

### 4.6. Cytokine Assays

Levels of IL-4, IL-10 and IL-17A in cell culture supernatants were measured using the Meso Scale Discovery^®^ (MSD) platform (MSD, Rockville, MD, USA). IFNγ in cell culture supernatants was measured by ELISA (Invitrogen). All assays were performed according to the manufacturers’ instructions.

### 4.7. Statistical Analyses

The Mann–Whitney U test was used to evaluate differences between two groups. For testing of differences between related samples Wilcoxon signed rank test (two groups) and Friedman’s test with Dunn’s multiple comparison test (several groups) were used. All statistical analyses were performed using GraphPad Prism 6.0 (GraphPad Software, La Jolla, CA, USA); *p*-values < 0.05 were considered as statistically significant. Data are shown as median (range) or median (inter quartile range, IQR) as defined in the text.

## Figures and Tables

**Figure 1 ijms-22-03084-f001:**
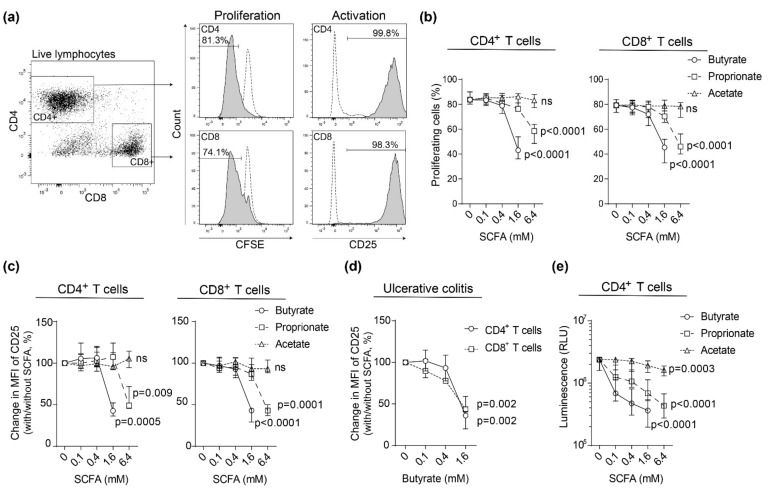
Effects of short-chain fatty acid (SCFA) on blood T cell proliferation and activation. Carboxyfluorescein succinimidyl ester (CFSE) labeled peripheral blood mononuclear cells (PBMCs) were stimulated with anti-CD3 and anti-CD28 with increasing concentrations of butyrate (0–1.6 mM), proprionate (0–6.4 mM) or acetate (0–6.4 mM) for 72 h and analyzed by flow cytometry. (**a**) Left, gating strategy using a healthy subject for CD4^+^ and CD8^+^ T cells from live lymphocytes. Right, gating strategy for proliferating cells (CFSE intensity) and activated cells (CD25). Dotted line shows cells cultured without anti-CD3 and anti-CD28. (**b**) The frequency of proliferating T cells and (**c**) change in median fluorescent intensity (MFI) of CD25 in stimulated T cells from healthy subjects (n = 9). (**d**) Change in CD25 MFI in stimulated T cells from ulcerative colitis (UC) patients with active disease (*n* = 7). (**e**) Histone deacetylase (HDAC) I and II activity in purified CD4^+^ T cells stimulated with anti-CD3 and anti-CD28 for 48h followed by addition of SCFAs at increasing doses for 1.5 h and subsequent HDAC-Glo^TM^ I/II assay (*n* = 6 healthy subjects). Data are shown as median and interquartile range.

**Figure 2 ijms-22-03084-f002:**
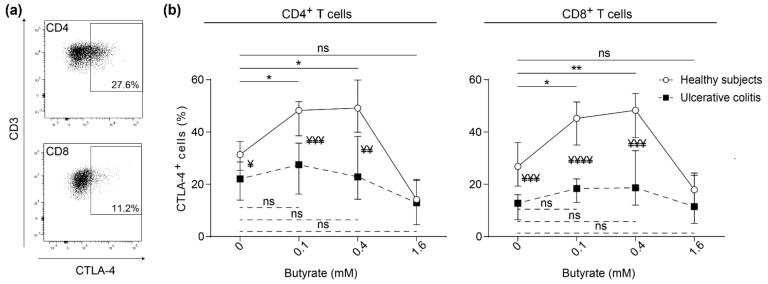
Effects of butyrate on cytotoxic T lymphocyte-associated antigen-4 (CTLA-4) surface expression on blood T cells from healthy subjects and UC patients with active disease. PBMCs were stimulated with anti-CD3 and anti-CD28 with increasing concentrations of butyrate (0–1.6 mM) for 72 h and analyzed by flow cytometry. (**a**) Gating strategy in an active UC patient for CTLA-4 expression on CD4^+^ (top) and CD8^+^ (bottom) T cells. (**b**) Frequency of CTLA-4 expressing CD4^+^ and CD8^+^ T cells among healthy subjects (open circles, *n* = 9) and patients with active ulcerative colitis (solid squares, *n* = 7). Comparisons within healthy subjects (solid lines) and within UC patients (dotted lines) are denoted with * where * < 0.05 and ** < 0.01. Comparisons between healthy subjects and UC are denoted with ¥ where ¥ < 0.05, ¥¥ < 0.01, ¥¥¥ < 0.001 and ¥¥¥¥ < 0.0001. Ns = non-significant. Data are shown as median and interquartile range.

**Figure 3 ijms-22-03084-f003:**
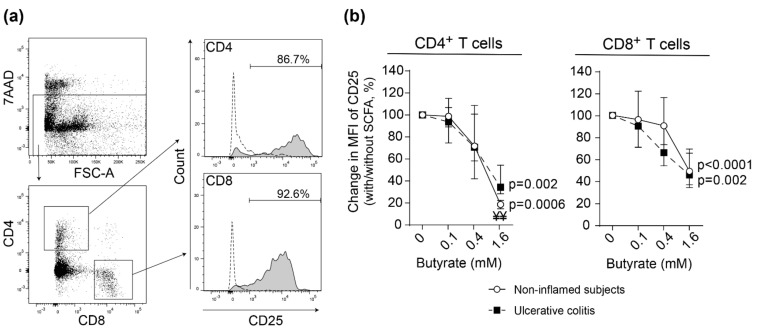
Effects of butyrate on lamina propria T cell proliferation and activation. Lamina propria (LP) cells were stimulated with anti-CD3 and anti-CD28 with increasing concentrations of butyrate (0–1.6 mM) for 48 h and analyzed by flow cytometry. (**a**) Gating strategy using LP cells from a non-inflamed subject. Live LP cells (7AAD^−^) were gated into CD4^+^ and CD8^+^ T cells (left panel) and further gated for activated cells (CD25). Dotted line shows cells cultured without anti-CD3 and anti-CD28. (**b**) Change in CD25 median fluorescent intensity (MFI) in stimulated CD4^+^ (**left**) and CD8^+^ (**right**) T cells from inflamed sigmoid colon of UC patients (*n* = 4) and sigmoid colon of non-inflamed subjects (*n* = 6). Data are shown as median and interquartile range. Comparisons between non-inflamed subjects and UC are denoted with ¥ where ¥¥ < 0.01.

**Figure 4 ijms-22-03084-f004:**
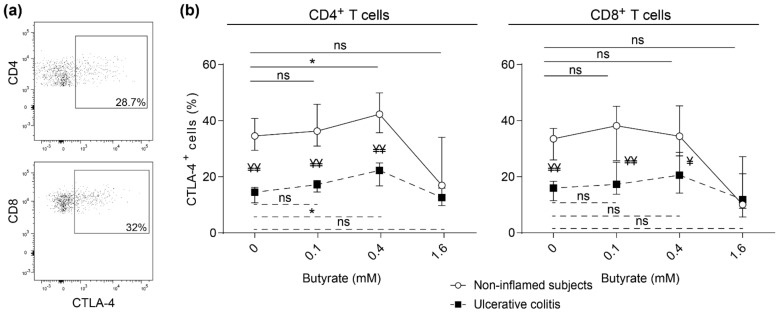
Effects of butyrate on CTLA-4 surface expression on lamina propria T cells from healthy subjects and UC patients with active disease. Lamina propria (LP) cells were stimulated with anti-CD3 and anti-CD28 with increasing concentrations of butyrate (0–1.6 mM) for 48h and analyzed by flow cytometry. (**a**) Gating strategy using a non-inflamed subject for CTLA-4 expression on CD4^+^ (top) and CD8^+^ (bottom) T cells. (**b**) Frequency of CTLA-4 expressing CD4^+^ and CD8^+^ LP T cells among non-inflamed subjects (open circles, *n* = 6) and patients with active ulcerative colitis (solid squares, *n* = 6). Comparisons within non-inflamed subjects (solid lines) and within UC patients (dotted lines) are denoted with * where * < 0.05. Comparisons between non-inflamed subjects and UC are denoted with ¥ where ¥ < 0.05 and ¥¥ < 0.01. Ns = non-significant. Data are shown as median and interquartile range.

**Figure 5 ijms-22-03084-f005:**
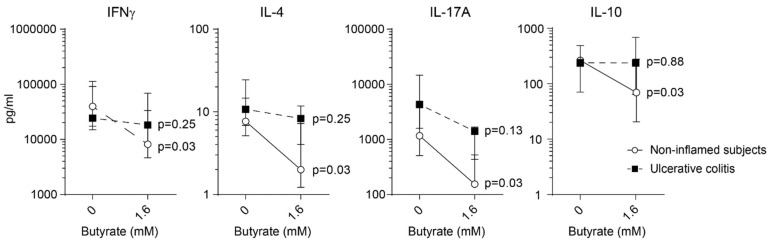
Effects of butyrate on cytokine expression in lamina propria T cells from healthy subjects and UC patients with active disease. Lamina propria (LP) cells were stimulated with anti-CD3 and anti-CD28 and treated with butyrate (1.6 Mm) for 48 h. Cytokine production of IFNγ, IL-4, IL-17A, and IL-10 were analyzed by ELISA and MSD. Data are shown as median and interquartile range.

## Data Availability

The data that support the findings of this study are available from the corresponding author upon reasonable request.

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
