# Peer review of "Impaired Butyrate Induced Regulation of T Cell Surface Expression of CTLA-4 in Patients with Ulcerative Colitis"

_ijms, 2021, doi:10.3390/ijms22063084_

Round 1
Reviewer 1 Report
The authors investigated the effect of butyrate on T cells and CTLA-4 expression in peripheral blood and colon lamina propria. They found that 1) butyrate decreased proliferation and activation of T cells in both healthy subjects and UC patients, 2) CTLA-4 less expressed in UC compared to normal, and 3) butyrate increased the expression of CTLA-4 more in healthy subject in peripheral blood. The manuscript is well-written. I have some comments as below.
Major comments:
- Blood T cells showed that butyrate did not upregulate CTLA-4 expression in both CD4+ and CD8+ T cells in UC patients in blood sample. However, LP T cells similarly stimulated CTLA-4 expression by butyrate in both normal subjects and UC patients. Please discuss this discordance.
- Figure 3. How did butyrate influence the cell proliferation in stimulated LP T cells?
- Please describe the histological findings of biopsies taken from inflamed and non-inflamed colon. Also, it would be better to describe where CD4+/CD25+ cells were located in the biopsies using immunohistochemistry.
Author Response
Point 1: Blood T cells showed that butyrate did not upregulate CTLA-4 expression in both CD4+ and CD8+ T cells in UC patients in blood sample. However, LP T cells similarly stimulated CTLA-4 expression by butyrate in both normal subjects and UC patients. Please discuss this discordance.
Author response: We agree that this is a discordance that needs to be discussed. Discussion concerning this has been added to page 6, lines 201-206.
Point 2: Figure 3. How did butyrate influence the cell proliferation in stimulated LP T cells?
Author response: The number of cells obtained from 6-8 biopsies for each patient were quite few and marking the cells with CFSE includes several steps of washing which each reduce the number of cells. Our focus was instead to have enough cells to study the dose-dependent effects of butyrate. Thus, we decided to omit effects on proliferation and only show effects on activation and CTLA-4. We acknowledge that there was an error concerning this in the abstract (page 1, line 22) and in the legend of Figure 3 (page 5, line 143) which have now been corrected. We only show activation, not proliferation.
Point 3: Please describe the histological findings of biopsies taken from inflamed and non-inflamed colon. Also, it would be better to describe where CD4+/CD25+ cells were located in the biopsies using immunohistochemistry.
Author response: We agree that histological evaluation of intestinal tissue as well as location of the CD4+/CD25+ cells in the biopsies would have been optimal, however, all available biopsies were used to obtain enough cells for the butyrate analysis. We have included this as a limitation of the study in the discussion on page 7, lines 248-250.
Reviewer 2 Report
An interesting original article describing butyrate's impaired ability to reduce cytokine secretion and induce surface expression of CTLA-4 in T cells from ulcerative colitis patients with active disease. Only minor queries:
I would probably dedicate some more space in the introduction to the description of Ulcerative condition as a clinical entity, instead of immediately focusing on short fatty acids .
Other than this comment, the paper is in my opinion well structured and coincise.
Thank You
Author Response
Point 1: I would probably dedicate some more space in the introduction to the description of Ulcerative condition as a clinical entity, instead of immediately focusing on short fatty acids.
Author response: We have now made a better description of ulcerative colitis at the beginning of the introduction (page 1, lines 35-42), and added a new reference.